# Common Peroneal Nerve Paralysis Following Rapid Weight Loss—A Case Report and Literature Review

**DOI:** 10.3390/nu17111782

**Published:** 2025-05-24

**Authors:** Laura-Elena Cucu, Gabriela Popescu, Alexandra Maștaleru, Emilian Bogdan Ignat, Cristina Grosu, Lenuța Bîrsanu, Maria Magdalena Leon

**Affiliations:** 1Faculty of Medicine, “Grigore T. Popa” University of Medicine and Pharmacy, 700115 Iași, Romania; dudau.laura-elena@d.umfiasi.ro (L.-E.C.); lenuta.balint-birsanu@d.umfiasi.ro (L.B.); 2Clinical Rehabilitation Hospital, 700661 Iași, Romania; alexandra.mastaleru@umfiasi.ro (A.M.); emilian.ignat@umfiasi.ro (E.B.I.); cristina.grosu@umfiasi.ro (C.G.); maria.leon@umfiasi.ro (M.M.L.); 3Departament of Medical Specialities I, “Grigore T. Popa” University of Medicine and Pharmacy, 700115 Iași, Romania; 4Department of Neurology, “Grigore T. Popa” University of Medicine and Pharmacy, 700115 Iași, Romania

**Keywords:** slimmer’s palsy, foot drop, rehabilitation therapy, peroneal neuropathy, weight loss, bariatric surgery

## Abstract

Common peroneal nerve neuropathy at the fibular head secondary to weight loss is known as slimmer’s paralysis. Although this pathology has long been documented in medical literature, it has gained more clinical significance in recent years due to the global rise in obesity and the increasing pursuit of rapid weight loss methods. While case reports exist in the current literature, there are limited data regarding its optimal management. This study summarizes all reported cases of common peroneal nerve paralysis after weight loss and reports one additional case, exploring disease mechanisms as well as diagnostic and therapeutic strategies. A literature review was conducted on the platforms PubMed, Google Scholar, and EMBASE. A total of 380 patients were included. Laterality of neuropathy was specified in 297 (78.16%) patients: 285 (95.96%) with unilateral neuropathy and 12 (4.04%) with bilateral neuropathy. A total of 19 (5.00%) patients had sudden onset, and in 145 (38.16%) of cases, the Tinel’s sign was positive. Additionally, 373 (98.16%) patients underwent nerve conduction studies, demonstrating the fibular head as the site of injury. MRI or ultrasound imaging of the knee is indicated to exclude compressive etiology. A total of 302 (79.47%) cases were treated surgically and 42 (11.58%) conservatively, predominantly with favorable outcomes, regardless of the therapeutic approach chosen. Although the predominance of surgically treated cases in the literature limits definitive treatment recommendations, conservative management appears appropriate when an extended recovery time is acceptable, while surgical decompression is indicated in cases showing no improvement after 3 months.

## 1. Introduction

According to the WHO definition, obesity is the excessive accumulation of fatty tissue, which, over time, leads to the development of pathologies with metabolic determinism. Obesity is on the rise as urbanization grows, with 18% of adults expected to have this disease by 2030. When it comes to the gender gap, it seems that women are more likely to be obese than men (27% vs. 23%). Although the global trend is for obesity to be more common among women, in Europe and the Western Pacific, the propensity for obesity is higher among men [1].

The most recommended and attainable strategies for weight loss include a low-calorie diet, increased physical activity, and lifestyle modifications to enhance weight maintenance post-weight loss. Nonetheless, despite the apparent simplicity of these methods, they are presently the least efficacious, as they are heavily contingent on the patient’s compliance with the new lifestyle. Consequently, patients may use weight loss medication alongside exercise for a more rapid and enduring effect [2,3].

The preferred therapy for morbid obesity continues to be bariatric surgery. The surgical approach leads to substantial weight reductions, consequently lowering cardiovascular risk and inducing remission of type 2 diabetes. Still, patients may experience long-term complications, such as iron deficiency, anemia, and deficiencies in vitamin B12 and folic acid, potentially leading to neurologic impairment (notably polyneuropathy) over time. Medical guidance post-surgery can effectively mitigate these deficiencies through oral supplementation [4]. In certain patients, vitamin B12 deficiency arises from malnutrition resulting from intake restriction, particularly due to a diminished intake of solid foods following surgery. Another cause is that post-surgery, the hydrochloric acid levels in the stomach are diminished, resulting in impaired protein digestion compared to a healthy individual, which leads to a deficiency in intrinsic factor and consequently, a vitamin B12 deficiency [5,6].

When it comes to bariatric surgery, it seems that the procedure is more common in women than men, as can be seen in a study published in 2018 that included 61,708 patients who chose this weight loss procedure. Of these, the highest percentage was found among women (78%) compared to men (22%) This difference occurs due to the much greater interest in weight on the part of women, as excess pounds have a significant emotional impact [7,8].

As the focus on weight management grows, awareness of potential complications from rapid weight loss requires careful consideration. In 1929, Woltman [9] was the inaugural author to document the correlation between significant weight loss and the presence of foot drop. In 1947, Kaminsky [10] and Denny-Brown [4] documented multiple instances of peroneal nerve palsy among prisoners of war, affecting up to 10% of the prisoners, and described this neurological disorder to extended dietary deprivation.

The peroneal nerve, which is one of the terminal branches of the sciatic nerve, is superficially located at the neck of the fibula, covered only by skin and subcutaneous tissue. Because of this anatomical position, it is susceptible to various lesions caused by pressure on the bone. Rapid and significant weight loss can lead to a condition called slimmer’s paralysis. With weight loss, the fatty tissue around the fibular head also decreases, therefore the nerve loses its protection against the bone [11]. This predisposes the nerve to compression, particularly during poor positions [12]. Since this nerve innervates the lower limb, its compression causes peroneal palsy, resulting in paresthesia, sensory deficits, and weakness in the extensors of the toes, ankle dorsiflexors, and foot evertors. The patient presents an abnormal gait, called a steppage gait, consisting of lifting the foot higher than usual when stepping forward to avoid hitting the toes to the ground [13].

Treatment can be conservative or surgical. Surgical decompression, although mainly documented in other causes of peroneal neuropathy, involves the dissection of fibrous elements covering the nerve at the fibular head, including the fibrous origin of the peroneus longus muscle, superficial fascia, and intermuscular septa [14].

## 2. Case Report

We present the case of a 35-year-old female smoker (8 pack year), with a history of bronchial asthma and elevated blood pressure who presented for cardiovascular evaluation 6 months following gastric sleeve surgery. The patient underwent a sleeve gastrectomy surgery in February 2024 in another medical institution. Her postoperative recovery was favorable, and she also received vitamin therapy and nutritional counselling. Before the surgery, the patient was 167 cm tall and weighed 104 kg (BMI = 37.2 kg/m^2^). The patient’s weight decreased linearly by approximately 42% over a six-month period. By the end of July, when the left lower limb motor deficit appeared, the patient’s weight had decreased to 63.2 kg (38% reduction from preoperative weight). At presentation (6.5 months after the surgery), she weighed 60.5 kg (BMI = 21.69 kg/m^2^). She reported a normalization in blood pressure values in recent months but complained about difficulty walking. She also described intermittent numbness on the lateral calf when crossing her legs or kneeling for long periods; these symptoms were observed several months before the bariatric surgery. These sensations disappeared completely if she changed her position and did not cause significant discomfort. Five months after the procedure, she noticed lower back pain and progressive weakness in her left leg. The symptoms progressed to her right leg after two weeks.

The patient’s past medical history included elevated blood pressure, chronic venous insufficiency CEAP C2, persistent bronchial asthma GINA step 3, and she is currently on treatment with Beclomethasone/Formoterol inhaler and oral Montelukast. No other potential cause for neuropathy was identified (diabetes, rheumatologic diseases, or toxic exposure). The patient stated that she is employed full-time as a shop assistant, and her sole daily physical activity consists of work-related tasks or household duties regarding her three children.

During the examination, we detected an important motor deficit in the lower limbs. The clinical assessment revealed bilateral steppage gait with foot drop due to severely compromised motor strength in the lower limbs. Voluntary movement was assessed with the Medical Research Council (MRC) [15] scale (ranging from 0 to 5, where 5 is normal). The patient was unable to perform ankle dorsiflexion (MRC 1/5 right, 0/5 left), big toe extension (MRC 0/5 bilaterally), and foot eversion (MRC 0/5 bilaterally), while foot inversion and plantar flexion were normal (MRC 5/5 bilaterally). The deep tendon reflexes were normal (2+ on the Deep Tendon Reflex Scale [16]). She had diminished sensation to light touch bilaterally in the ventral and lateral region of the calf and on the dorsal surface of the foot. The Tinnel’s sign was positive bilaterally.

Considering her past medical history and physical findings, peroneal nerve damage due to significant weight loss was suspected. Lumbar MRI, performed to rule out spinal nerve compression, showed disc protrusions at L4-L5 and L5-S1 levels without nerve root involvement (Figure 1). Neurosurgical and electrophysiological assessments further confirmed the absence of spinal nerve compression as the cause of the motor deficit. Nerve conduction studies bilaterally showed focal neuropathy of the common peroneal nerve localized around the fibular head. Involvement was more severe on the left, with axonal loss (Figure 2) and reduced sensory nerve action potential amplitude in the superficial peroneal nerve.

Knee ultrasonography demonstrated inflammation of the nerve adjacent to the fibular head and neck, but without any evidence of extrinsic compression (Figure 3). To exclude other potential causes of neuropathy, a comprehensive panel of laboratory tests was conducted (Table 1). No significant deficiencies in vitamin B12, folate, or iron were observed at the first evaluation, even though the laboratory analysis showed a value below the normal laboratory limits for hemoglobin and erythrocytes. The patient thus received postoperative iron and folic acid therapy (100 mg/5 mL—one vial twice a day), injectable B12 (1000 micrograms/mL), and a vitamin complex in capsule form containing minerals (Mg, Ca, I, Cu, Mn, Zn, and Se), vitamins (K, D, C, A, B1, B2, B3, B6, B12, folic acid, and pantothenic acid, and extracts of Siberian ginseng, Spirulina, and Chlorella). Four months after the first evaluation, the hemoglobin value normalized, and the erythrocyte value increased. Due to financial reasons, vitamin B12, folate, and vitamin D are not routinely measured in our clinic, and the patient refused to evaluate these laboratory parameters.

The patient was diagnosed with bilateral common peroneal nerve palsy and treated with vitamin therapy, kinesitherapy, and transcutaneous electrical nerve stimulation (20 mA intensity for 10 min on each limb, using a BTL-500 device, BTL Industries Ltd., 2014, Stevenage, UK). She continued rehabilitation therapy throughout the 4-month period while maintaining a constant weight. At the follow-up evaluation, an almost complete recovery was observed, with significant improvement in muscle strength for ankle dorsiflexion (MRC 4/5 bilaterally), big toe extension (MRC 4/5 on the right, MRC 3/5 on the left), and foot eversion (4/5 bilaterally) (Appendix A). There was also an improvement in sensory disturbances and enhancement of both sensory and motor electrophysiological parameters bilaterally (Appendix A).

## 3. Literature Review

### 3.1. Materials and Methods

In order to observe the impact of rapid weight loss on the peripheral nervous system in more detail, we conducted a literature review using the words “slimmer’s paralysis”, “bariatric surgery”, “rapid weight loss”, and “slimmer’s palsy” on Google Scholar, PubMed, and EMBASE. The results can be seen in Figure 4.

Inclusion criteria: We have included articles presenting patients over 18 years of age who developed peroneal nerve paralysis following weight loss. The publication period was 2000–2025, including only articles written in English on Google Scholar, PubMed, and EMBASE.

Exclusion criteria: We have excluded conference papers, conference abstracts, conference posters, systematic reviews, meta-analyses, and literature reviews.

The article selection process for this research was conducted by two independent reviewers who initially assessed the titles of the identified publications. Subsequently, the abstract of each article was examined, and ultimately, a comprehensive reading of the relevant articles was performed for the final selection. In instances of uncertainty, the third reviewer intervened in the decision-making process.

### 3.2. Results

We have included 20 [11,17,18,19,20,21,22,23,24,25,26,27,28,29,30,31,32,33,34,35] articles published between 2000 and 2025 (Appendix A), consisting of nine case reports [17,19,21,22,24,29,32,34,35], five case series [11,20,23,31,33], five retrospective series [18,25,26,30,36], and one observational case-control study [27]. These publications originated from various countries: one each from Israel [32], Greece [31], Malaysia [21], South Korea [24], Brazil [23], and New Zealand [35], five from the USA [11,17,19,20,29], four from Turkey [18,22,26,34], three from Belgium [27,30,36], and two from Spain [25,33]. The 20 publications encompassed a total of 380 patients, with 252 (66.32%) being female, ranging from 22 to 79 years (mean age 50.87 years). In eight of the twenty articles, the participants lost weight through diet [11,21,25,27,29,30,32,36] and in two of them, they also included physical activity [11,29]. In two articles, the participants experienced weight loss after receiving a GLP-1 agonist [19] and Tirzepatide treatment [20]. Complications following bariatric surgery were reported in 10 articles, comprising 86 patients (22.63%), with detailed patient characteristics presented in Appendix A.

## 4. Discussions

The advancement of medicine has led to the development of bariatric surgery options. Neurological complications following bariatric surgery have been reported in 1.18–16% of cases [17,37,38,39]. Depending on their onset, these complications can be acute (within the first few months post-surgery), including polyradiculoneuropathy, rhabdomyolysis, and Wernicke’s encephalopathy; subacute (1.5–3 years post-surgery), such as optic neuropathy; and chronic (3 years after surgery), including myelopathy, peripheral neuropathies, and myopathy [37,40,41]. Mononeuropathies are acute/subacute complications, occurring in the first month [18], to 1–1.5 years after surgery [37,39,40,42].

Most of these complications achieve at least a partial recovery following vitamin supplementation, being generally linked to vitamin and mineral deficiencies [36,37,41]. This theory is supported by Thaisetthawatkul et al. [38], who compared two matched groups of patients, one who underwent cholecystectomies and one with bariatric surgery, with no significant differences in body weight before and after surgery. Peripheral neuropathies and mononeuropathies were more frequent in the bariatric surgery group than in the cholecystectomy group (16% vs. 3%) and the authors attributed this to the surgical procedure and secondary malabsorption. Even if patients with these complications have normal tests, a deficiency cannot be ruled out completely, because the serum values cannot determine the presence of an eventual imbalance in the nervous tissue.

### 4.1. Risk Factors

The patients experienced weight loss of 15 to 50 kg over approximately six months, with an average of 5.4 kg per month (38.11–51.81% of their initial weight in the first 6 months and 40% of their initial weight in the first year). The case we reported corresponds to the same patient profile, involving a 35-year-old woman who lost 42% of her initial body weight within 6 months, with the onset of symptoms occurring approximately 5 months post-surgery. The amount and rapidity of weight loss, especially in the first 6 months after surgery, is a controversial risk factor. Some authors consider there to be a link with the onset of paralysis [18,19,21,22,27,28,36], while others do not agree [26].

Among the twenty articles included, the patient’s smoking status is mentioned in four of them [11,19,34,35] and the alcohol consumption in another four—3 patients (0.79%) were not consuming [29,34,35] and 2 patients (0.53%) had alcohol abuse [31]; in the rest, this information is missing. Of the total number of patients, 7 (1.84%) were diagnosed with diabetes [11,17,20,31,34], 1 (0.26%) with impaired fasting glucose [20], and 19 (5.00%) of them did not have the disease [11,19,21,22,23,29,32,33,35]. For the others, the presence or absence of diabetes is not specified. Diabetes and smoking are among the possible risk factors [36,43], but in many of the patients in the review, these data are missing. The presence of polyneuropathy was proposed as a predisposing factor [17].

Maintaining poor postures, such as leg crossing, prolonged sitting, or squatting, was reported in 34 (8.95%) cases [11,19,20,21,22,25,29,31,33,34], but this low percentage is underestimated because most of the patients in the review came from large studies [18,26,28,30]. Immobility was assumed as a risk factor in 7 (1.84%) cases [11,24,30,31], with one of them (0.26%) unable to move for two weeks following biliary surgery [24]. Minor knee trauma was identified as a risk factor in one (0.26%) case [35]. Walking too much (10.000–15.000 steps per day) was also mentioned in one (0.26%) case [29] and effort was mentioned in another four (1.05%) cases [31]. Table 2 summarizes the risk factors for peroneal nerve paralysis following rapid weight loss.

### 4.2. Proposed Pathophysiological Mechanisms

The proposed causes of peroneal nerve paralysis are summarized in Figure 5. Most of the articles included in the review, except for two that did not mention any mechanism [23,30], support the mechanical hypothesis—after significant weight loss, the fat pad and the subcutaneous adipose tissue are lost and the nerve is subsequently compressed on the bone. These changes are also observed on knee MRI [29,30,32]. One study suggested that even the perineural and intraneural fat is lost as well [27], while others support the existence of a microcirculatory nerve dysfunctions with ischemia [11,17,25] and/or edema [11,17,27,28]. Some authors, who reported cases of bilateral common peroneal nerve paralysis, believed that the mechanism would be an inflammatory one [17,22,35], based on the biopsy results of some patients with polyneuropathy [41]. In another article, the authors suggested that, in the case of bilateral damage, there could be a systemic cause [32]. However, there may be a connection between all these phenomena. Metabolic and hormonal alterations secondary to weight loss [11,21,22,24] could increase the nerve’s susceptibility to compression and mechanical stress can cause edema, inflammation, and ischemia, leading to decreased nerve conduction and paralysis. Fibrous elements [11,17,26] and anatomical variants (such as an upper bifurcation or a more superficial trajectory of the nerve) [11] can also contribute to mechanical compression. In addition, Maylaerts et al. [27] found a positive relationship between BMI and fibular tunnel size, as well as with the thickness of the nerve.

### 4.3. Clinical Manifestation

Sudden onset of symptoms was reported in 19 (5.00%) patients [18,19,21,26,35]. One of these cases (0.26%) suffered from foot drop following prolonged exposure to the reverse Trendelenburg position during surgery [18] and another diabetic patient (0.26%) had bilateral foot drop after 4 days of ventilation and sedation [11]. The Tinel’s sign was positive in 145 (38.16%) cases [24,29,30,32], negative in 59 (15.53%) [17,30], and not mentioned in the remaining 176 (46.32%) patients.

The onset of the disease with back pain in our patient is an observation lacking a clear interpretation. It was also described in two other cases [31,32], one of them with severe spinal stenosis [32]. Considering the limited number of patients with this complaint, it could be an incidental association, because back pain is frequent in the general population as well.

First, clinicians should observe if the patient demonstrates a steppage gait. Then, they should perform an inspection and palpation to identify external compression, check for Tinel’s sign, and follow this with an assessment of muscular strength and sensory disturbances. Weakness in foot eversion and diminished sensation in the lower lateral leg and dorsum of the foot suggest superficial peroneal nerve (SPN) involvement, while impaired foot and toe dorsiflexion alongside sensory deficits in the area between the first and second toes indicate deep peroneal nerve (DPN) involvement [44]. Both are branches of the common peroneal nerve.

For 83 (21.96%) patients, we do not have data on whether the damage was unilateral or bilateral. Among the other 297 (78.16%), 285 (95.96%) presented with unilateral foot drop—132 (46.32%) in the right leg, 116 (40.70%) in the left one, and for 37 (12.98%), it was not specified. A total of 12 (4.04%) patients had bilateral foot drop. The symptoms became bilateral at the same time in two (16.67%) cases [11,18], during variable intervals (1–6 months) in five (41.67%) cases [17,19,22,32], and in the rest of the cases, the time interval was not mentioned [21,23,24,29,31]. Our case is notable, as bilateral involvement was observed in only a small minority of patients from our review.

### 4.4. Investigations

Before surgery, patients need to have a thorough check-up that includes standard tests like a complete blood count, metabolic panel, fasting blood sugar, hemoglobin A1c, lipid panel, urinalysis, and blood clotting tests, along with checks for vitamin B12, folic acid, iron, and vitamin D. Moreover, before the surgical intervention, we should examine patients for possible malabsorption syndromes to prevent any deficiency [45]. In addition, cardiological and pulmonary evaluations are mandatory for assessing intraoperative and postoperative risk. Moreover, to evaluate the pre-operative risk, cardiopulmonary testing is useful for quantifying the level of effort that the patient will be able to perform after the surgical intervention; in addition to dietary changes, the patient will also receive exercise recommendations. Studies indicate that even a gastroenterological evaluation, including endoscopy, is necessary before bariatric surgery to determine the condition of the organ that will be surgically intervened. Gastroesophageal reflux disease, Barrett’s esophagus, or *H. pylori* infection can impede making the bariatric decision [46,47,48]. Moreover, patients must be evaluated by an endocrinologist to rule out any endocrine disorders or to initiate appropriate therapy that does not affect postoperative recovery [45,49].

Considering the physical and psychological impact that a surgical intervention and weight loss have on patients, continuous collaboration both before and after the surgical intervention is essential [50,51,52]. Patients must be evaluated to rule out eating disorders, considering that after the surgical intervention, a restrictive diet will be imposed. In addition, a comprehensive dental examination is necessary to avoid complications that may arise due to dental issues. Post-operatively, patients must chew well and easily for optimal digestion and to avoid vomiting, abdominal pain, diarrhea, and dumping syndrome. Thus, the texture of the foods must be adapted according to the dentition [53,54].

The investigations used to confirm the diagnosis are biological analyses (including the usual risk factors for neuropathies), electrophysiological investigations (nerve conduction studies and electromyography) and imaging studies (lumbar MRI, knee MRI, and knee ultrasound). Imagistic assessments are useful to exclude other external compressive causes, such as a Baker’s cyst, post-traumatic hematoma, lipoma, or other benign or malignant tumors. Although both ultrasonography (US) and MRI are utilized, US is more frequently used due to its accessibility.

When discussing blood tests, in 14 of the 20 articles included in the review, details related to this aspect are presented. Researchers conducted blood tests to assess vitamin B complex, vitamin D, diabetes, kidney and liver function, calcium, magnesium, and iron levels. One paper assessed the patients’ lipid profile [22], along with zinc, selenium, copper, retinol, and protein levels [11], while another article examined antinuclear antibodies (ANAs), complement levels, and anti-double-stranded DNA [20]. Albumin levels were evaluated in two studies [11,35]. Thyroid function was evaluated in six (1.58%) cases [20,21,24]. Genetic tests were performed in three cases—two (0.53%) for neuropathies [25,29] and one (0.26%) for neuromuscular disorders [29]. Most laboratory values were normal, with three exceptions: elevated copper in one (0.26%) case [20], low vitamin B6 in one (0.26%) case [20], and borderline normocytic anemia in one (0.26%) case [35].

Nerve conduction studies and electromyography were performed in 373 (98.16%) patients, finding demyelination with a partial or complete conduction block of the peroneal nerve at the fibular head, with only one case lacking an electrophysiological evaluation [23]. Other pathological changes were axonal loss [11,22,31], abnormal spontaneous activity in the tibialis anterior, peroneus longus, or extensor digitorum brevis [17,21,22,24,25,29,31,35], absent or reduced sensory nerve action potentials (SNAPs) in the superficial peroneal nerve [11,17,22,24], and polyneuropathy [11,17,20,25,30]. Cruz-Martinez et al. [25] analyzed the evolution of 30 patients with acute unilateral common peroneal neuropathy. They focused on a differential diagnosis with hereditary neuropathy with pressure palsies (HNPPs), a disease that can cause sensory or motor mononeuropathies in young people. The authors recommended comprehensive electrophysiologic studies to exclude polyneuropathies and suggested that a conduction block could be secondary to ischemic or metabolic changes due to compression. They also found a correlation between the severity of the conduction block, the clinical presentation, and the recovery time. Axonal loss can also be found on nerve conduction studies, and its degree is believed to be correlated with the prognosis [14,31]. This observation is consistent with our experience with other mononeuropathies, but among the cases we have reviewed, there were patients with complete recovery only with conservative treatment, even though axonal loss or denervation in common peroneal nerve-innervated muscles was present [11,21,29,33].

The knee ultrasound was performed in 240 (63.16%) patients, showing an enlarged common peroneal nerve at the fibular neck [11,34], slightly increased neural vascularity [20], hypoechogenity [27], or a decrease in subcutaneous fat [27].

In some studies, an MRI of the knee was performed only in those with ultrasound abnormalities [26,30], while other authors used it as an initial evaluation [17,19,29,32]. The pathological changes found were abnormal T2 hyperintensity (edema) of the nerve at the fibular head in 13 (3.42%) cases [11,29,30,34], three (0.79%) patients with thinning of the subcutaneous fat and soft tissue near the fibular head [29,30,32], one (0.26%) patient with compression [19], and one (0.26%) with neuritis of the peroneal nerve [30]. There were also 23 (6.05%) cases without clear pathological changes [17,30]. In addition, atrophy of the anterior tibialis muscle was described in 16 (4.21%) patients [17,30]. A more extensive pattern of atrophy was described in other cases: anterior compartment muscles in two (0.56%) patients [17,29] and both anterior and lateral compartment muscles in one (0.26%) case [24].

Other investigations conducted included lumbar MRI [11,17,24,26,30], lumbar CT [32], and knee X-rays [19].

### 4.5. Treatment

There were two types of treatment, with surgical intervention carried out in 302 (79.47%) cases and conservative treatment in 42 (11.58%) patients. The remaining 34 (8.95%) patients either did not receive any treatment [37] or it was not mentioned [23,26,31]. The conservative treatment consisted of a balanced diet, vitamin and mineral supplements, avoidance of poor postures, physical therapy, ankle–foot orthosis, and rehabilitation therapy in some cases. Even though diet cannot replace electrophysiological therapy, rehabilitation, or even surgery, it is an essential pillar in the rapid recovery of these patients. Studies show that a diet rich in PUFAs ω-3 and ω-6, vitamins D, B9, and B12, and Mg^2+^, zinc, and Ca^2+^ is the most suitable for nerve regeneration [55]. In addition to a diet rich in nutrients and calories, it is essential that patients have as few gastrointestinal symptoms as possible and, likewise, as few postoperative complications that could lead to nutritional deficiencies. For this reason, a preoperative evaluation is essential [56].

Corticosteroid therapy was also tried in a limited number of patients, with clinical improvement in one patient after a week of methylprednisolone [11], but no results for another one after 4 days of dexamethasone [32]. In one case, vitamin B12 supplementation was administered despite the absence of a deficiency [35], with subsequent clinical improvement noted.

The preferred treatment in our review was surgical intervention, as two large studies (comprising 278 patients) [30,36] focused exclusively on surgically treated cases, without specifying selection criteria. In two articles, surgery was recommended when common peroneal nerve neuropathy had an acute onset, was severe, or was persistent [11,17]. However, 31 patients were treated conservatively even though they had sudden onset, and 29 of them recovered completely in 3 months [25].

Broekx and Weyns [30] treated 200 patients with external neurolysis and suggested that the surgery should be done as soon as possible after the onset of symptoms for better results. In their research, the average time from the onset of symptoms to the operation was 4 months. The mean period for maximal recovery was 3 months, and 4.5% of patients had postoperative complications. The authors found an association between the degree of muscle strength impairment and the surgical result. Lale et al. [26] analyzed the evolution of nine patients treated with peroneal nerve decompression and recommended early surgery. None of their cases had postoperative complications. The duration of the symptoms before the surgery was less than a month and a complete recovery was noticed in all patients in a maximum of 2 months.

Among the reviewed cases, the prognosis was generally good, regardless of the chosen treatment, although after surgical treatment the recovery was faster (2 weeks to 3 months [11,17,30]) than after conservative treatment (1 month [19] to 1.5 years [35]). In our case, the patient showed substantial though incomplete recovery after 4 months of conservative treatment. There is also one case with complete recovery without any treatment in 45 days [23]. We believe that the choice of treatment depended particularly on the experience of the center. Dwivedi et al. [14] wrote a review about foot drop management for all causes and recommend clinical reassessment at 4–6 weeks and electrophysiological follow-up at 6–12 weeks to check for recovery in patients who were conservatively treated. They recommend surgery if there was no clinical or electrophysiological improvement after 3–6 months. Another review on peroneal nerve palsy, written by Poage et al. [44] also suggested conservative treatment as the first-line approach and surgical decompression if there is no improvement after 3 months.

Considering the favorable clinical evolution of the patients in the review, even those with an acute onset or with signs of active denervation on EMG/NCS, we consider it appropriate to start with conservative treatment if a longer recovery time is acceptable for the patient. In our review, four cases without significant muscle strength improvement after conservative treatment were treated surgically, with two after 2 weeks [26] and two after 6 months [25,32]. However, with conservative treatment typically requiring longer recovery times, the surgery may have been premature, at least in the first two cases. Future studies should investigate whether structured rehabilitation programs could potentially reduce recovery time in conservatively managed patients. Therapeutic approaches to peroneal nerve paralysis are presented in Figure 6.

### 4.6. Study Limitations

The limitations of this study include data heterogeneity primarily concerning predisposing factors, attributable to the inconsistent reporting across studies, the absence of standardized follow-up periods, and the undefined criteria for partial recovery. A significant bias emerges from the overrepresentation of surgical treatment, as two large studies focused exclusively on these cases. Furthermore, the precise definition of conservative management remains ambiguous, with some patients experiencing spontaneous recovery, while others received vitamin supplementation, underwent rehabilitation programs, or utilized orthotic devices.

## 5. Conclusions

In conclusion, excessive weight loss can lead to common peroneal nerve paralysis, a complication to which both local and systemic factors contribute. Our review demonstrates that this condition has favorable outcomes with conservative management, even in acute presentations. While surgical decompression remains important for refractory cases, the decision between immediate surgical intervention versus conservative management should carefully balance recovery timeline expectations with surgical risks. This is particularly important as the predictors of treatment response remain poorly defined in the current literature.

## Figures and Tables

**Figure 1 nutrients-17-01782-f001:**
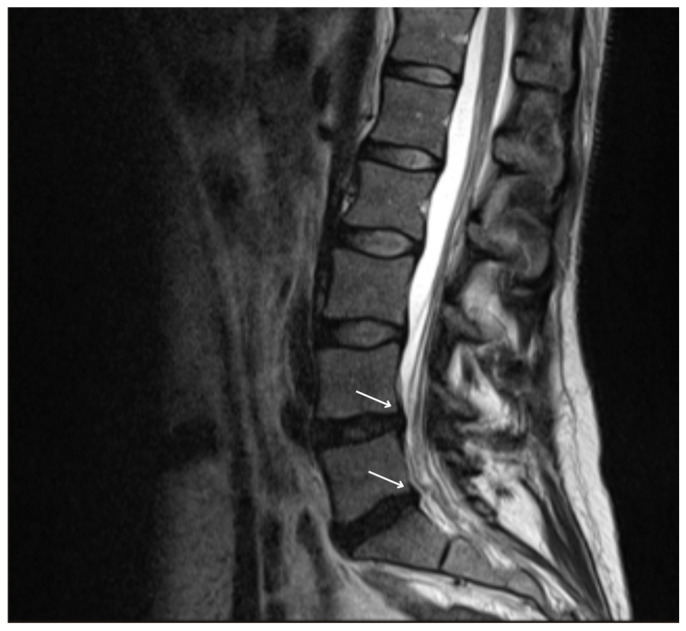
MRI of lumbar spine—sagittal T2 TSE. Disc protrusions at L4-L5 and L5-S1 levels (indicated by arrows).

**Figure 2 nutrients-17-01782-f002:**
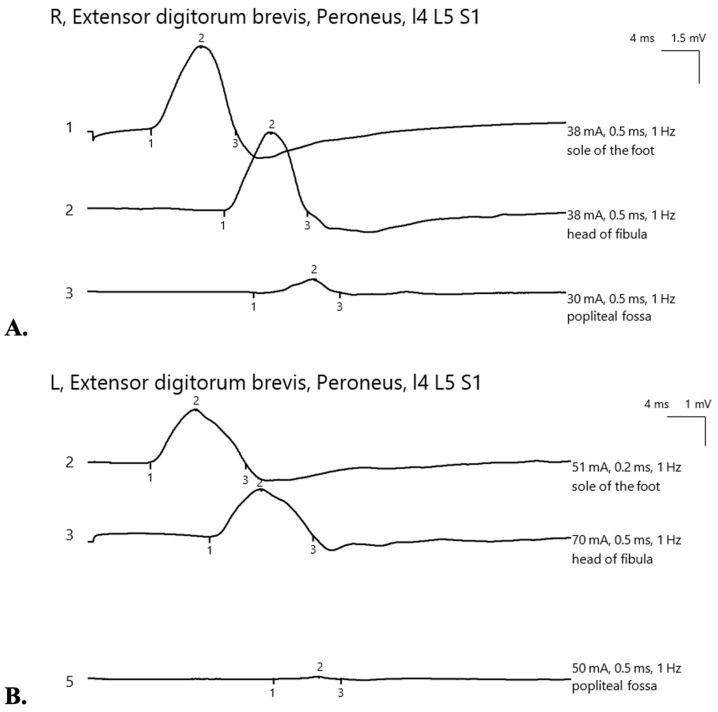
Nerve conduction study. (**A**). Right leg—conduction block of peroneal nerve at fibular head. (**B**). Left leg—reduced motor potential amplitude of peroneal nerve (suggesting axonal loss), with conduction block at fibular head.

**Figure 3 nutrients-17-01782-f003:**
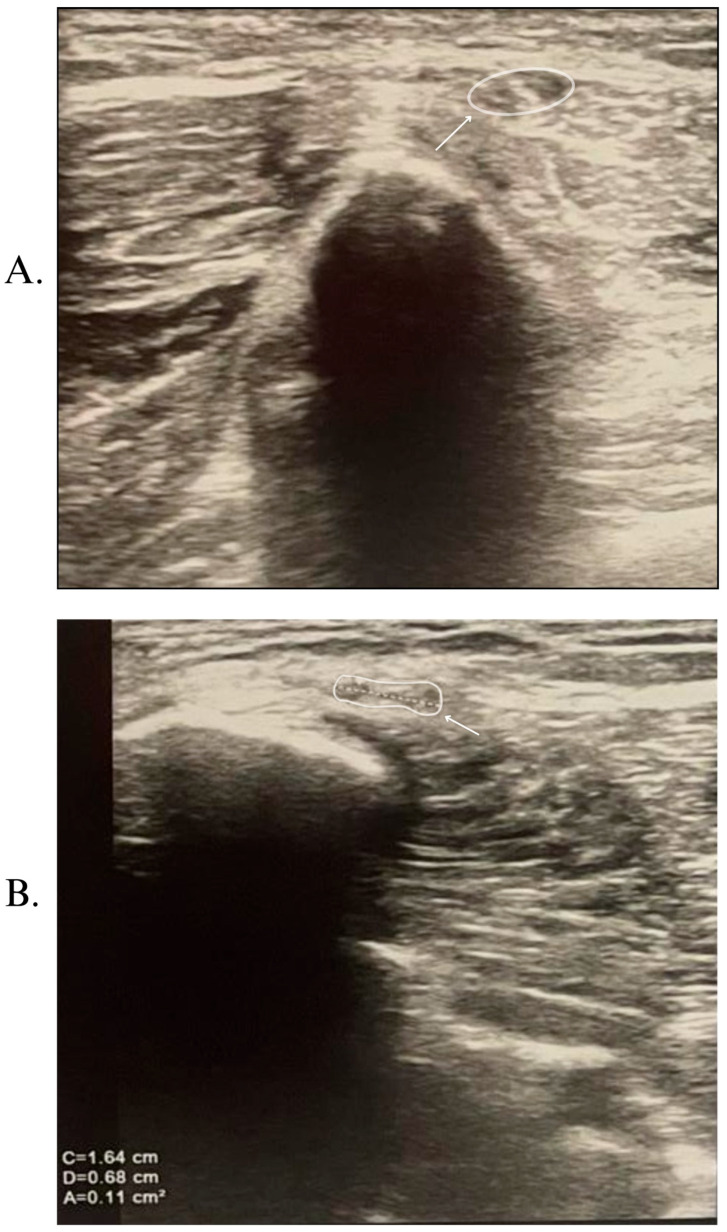
The ultrasound of the common peroneal nerve at the fibular head level. (**A**). Right and (**B**). left. The nerve structure is hypoechoic with a slightly enlarged cross-sectional area. The nerve is indicated by arrows, the surface is outlined by highlighted areas, and the dashed lines demonstrate the plane for the cross-sectional area assessment.

**Figure 4 nutrients-17-01782-f004:**
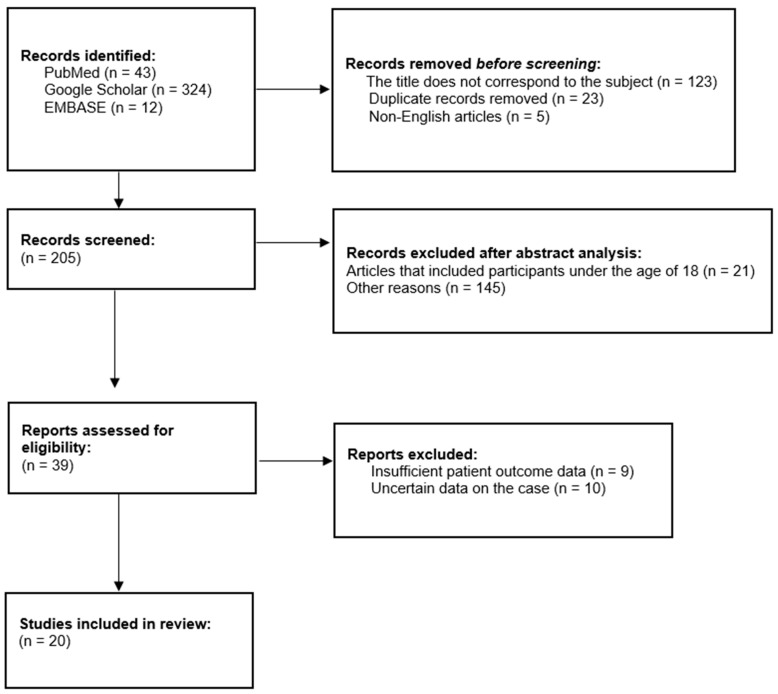
Graphical representation of process of evaluation and inclusion of articles in review.

**Figure 5 nutrients-17-01782-f005:**
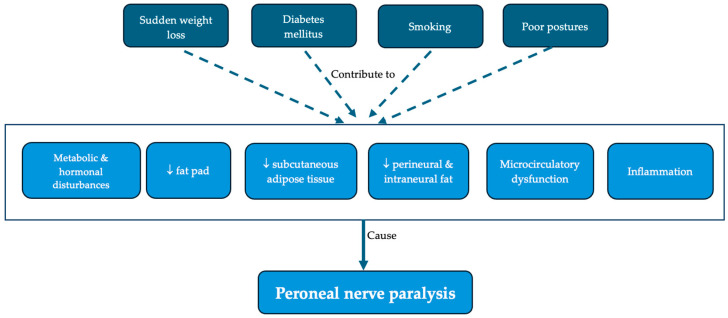
Proposed causes of peroneal nerve paralysis after rapid weight loss.

**Figure 6 nutrients-17-01782-f006:**
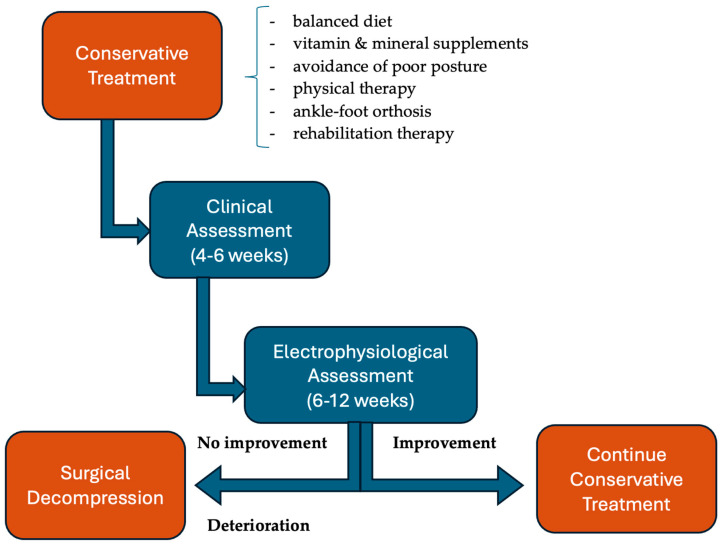
Approaches to treatment of peroneal nerve paralysis following rapid weight loss.

**Table 1 nutrients-17-01782-t001:** Blood tests. Abbreviations: WBC—White Blood Cells, RBC—Red Blood Cells, Hb—Hemoglobin, Ht—Hematocrit, MCV—Mean Corpuscular Volume, MCH—Mean Corpuscular Hemoglobin, MCHC—Mean Corpuscular Hemoglobin Concentration, RDW-CV—Red Cell Distribution Width-Coefficient of Variation, PLT—Platelets, VSH—Erythrocyte Sedimentation Rate, CRP—C-Reactive Protein, A1c%—Glycated Hemoglobin, TSH—Thyroid Stimulating Hormone, fT4—Free Thyroxine, and fT3—Free Triiodothyronine.

Parameter	Pre-Treatment Value	4-Month Follow-Up Value	Reference Range
WBC	6.78 × 10^3^/mm^3^	4.82 × 10^3^/mm^3^	4.00–10.00 × 10^3^/mm^3^
RBC	3.67 × 10^3^/mm^3^	3.79 × 10^3^/mm^3^	3.80–5.80 × 10^3^/mm^3^
Hb	11.9 g/dL	12.1 g/dL	12.0–16.0 g/dL
Ht	37.0%	37.6%	36.0–48.0%
MCV	100.8 fL	99.1 fL	78–100 fL
MCH	32.5 pg	32.0 pg	25.0–32.0 pg
MCHC	32.2 g/dL	32.2 g/dL	32.0–35.0 g/dL
RDW-CV	12.2%	11.8%	12–18%
PLT	334 × 10^3^/mm^3^	284 × 10^3^/mm^3^	150–400 × 10^3^/mm^3^
VSH	11 mm/h	9 mm/h	0–20 mm/h
CRP	0.01 mg/dL	0.28 mg/dL	0–1 mg/dL
Sodium	139.10 mmol/L	142.29 mmol/L	135–145 mmol/L
Potassium	3.83 mmol/L	4.43 mmol/L	3.5–5 mmol/L
Magnesium	2.01 mg/dL	1.91 mg/dL	1.6–2.6 mg/dL
Calcium (Total)	9.64 mg/dL	8.76 mg/dL	8.4–10.2 mg/dL
Ionized Calcium	4.96 mg/dL	4.01 mg/dL	3.82–4.82 mg/dL
Iron	111.2 μg/dL	-	50–170 μg/dL
Glucose	100.8 mg/dL	76.38 mg/dL	70–105 mg/dL
A1c%	4.71%	-	4–6%
TSH	0.537 μIU/mL	-	0.55–4.78 μIU/mL
fT4	1.24 ng/dL	-	0.89–1.76 ng/dL
fT3	3.03 pg/mL	-	2.3–4.2 pg/mL
Folate	8.01 ng/dL	-	>5.38 ng/dL
Vitamin B12	386 pg/mL	-	211–911 pg/mL
25-OH-Vitamin D	20.9 ng/dL	-	20–50 ng/dL

**Table 2 nutrients-17-01782-t002:** Risk factors for peroneal nerve paralysis following rapid weight loss.

Commonly identified risk factors	Diabetes [11,17,20,31,34,36,43]
	Smoking [11,19,34,35,36,43]
	Maintaining poor postures (leg crossing, prolonged sitting, squatting) [11,19,20,21,22,25,29,31,33,34]
	Immobility [11,24,30,31]
Controversial risk factors	Level and rapidity of weight reduction [18,19,21,22,27,28,36]
Risk factors requiring further research	Alcohol consumption [31]
	Polyneuropathy [17]
	Minor knee trauma [35]
	Excessive walking [29]
	Effort [31]

## Data Availability

The original contributions presented in the study are included in the article/Appendix A, further inquiries can be directed to the corresponding author.

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
