# Peer review of "Common Peroneal Nerve Paralysis Following Rapid Weight Loss—A Case Report and Literature Review"

_nutrients, 2025, doi:10.3390/nu17111782_

Round 1
Reviewer 1 Report
Comments and Suggestions for Authors
One of the significant problems with this manuscript is that it represents an attempt at presenting a case report along with a systematic review of the relevant literature on similar cases-ie patients developing peroneal nerve palsy following bariatric surgery.
However, the evidence base, which forms the basis of this systematic review is weak-ie it involves predominantly case reports/case series along with several retrospective reports of the management of these types of patients including with surgery in the largest series. Hence there is also a reasonable amount of bias as to what is being reported, by whom and when, noting that there is a lack of intention to treat data. Hence the fact that no formal assessment of the risk of bias has been undertaken as part of the systematic review is problematic, along with a failure to adequately summarize the level of evidence (Grading of the evidence for each manuscript).
All of the above seriously constrains any conclusions that can be drawn such that this section in the manuscript requires reconsideration. In addition, there are some other specific comments with respect to the content of the manuscript which include-
1) If the authors are going to posit that obesity is more common in females versus males (as per the first paragraph of the Introduction section), then they need to provide a lot more in the way of supporting information over and above one reference (including data, region of the world, time period etc).
2) Mention is made of electrical stimulation being used as part of the management in some cases. What did this involve?
3) Similarly what type of surgical procedures were being performed? Was it the same surgical procedure across all of the selected studies?
4) It would be useful to summarize the demographics etc of all of the patients who developed peroneal nerve palsy following bariatric surgery into one Table as a subgroup analysis. This would allow a more meaningful comparison between the case report and that particular subset of data. Of the 200 patients in the largest series it is evident that not all of these patients developed peroneal palsy following bariatric surgery
5) Is it known how many patients failed conservative therapy prior to proceeding to surgery?
6) The Table encompassing the laboratory results for the case that is being reported seems redundant in the body of the main manuscript and is best swapped with Table 2.
7) Table 2 does need some work, in that it needs to be made clearer how many patients developed peroneal nerve palsy following BS within the larger patient cohorts -ie for references 25 and 28. This is over and above what has been stated in point number 4 above.
Author Response
Dear reviewer, first of all, we would like to thank you for all your comments and suggestions! We believe that due to these suggestions and comments, the quality of our article has increased considerably! The answers and suggested modifications are provided in the main document.
- If the authors are going to posit that obesity is more common in females versus males (as per the first paragraph of the Introduction section), then they need to provide a lot more in the way of supporting information over and above one reference (including data, region of the world, time period etc).
We have added more details about this in the introduction.
- Mention is made of electrical stimulation being used as part of the management in some cases. What did this involve?
Regarding our case, we add more information about electrical stimulation (line 155-158).
- Similarly what type of surgical procedures were being performed? Was it the same surgical procedure across all of the selected studies?
We have added more details regarding these aspects in Table S2 in the supplementary data section.
- It would be useful to summarize the demographics etc of all of the patients who developed peroneal nerve palsy following bariatric surgery into one Table as a subgroup analysis. This would allow a more meaningful comparison between the case report and that particular subset of data. Of the 200 patients in the largest series it is evident that not all of these patients developed peroneal palsy following bariatric surgery
We have added more details regarding these aspects in Table S2 in the supplementary data section.
- Is it known how many patients failed conservative therapy prior to proceeding to surgery?
It appears that no such information is mentioned in the articles included in the study.
- The Table encompassing the laboratory results for the case that is being reported seems redundant in the body of the main manuscript and is best swapped with Table 2.
We really appreciate your suggestion, but we think it would be beneficial to keep the table in the case report section to emphasise the patient's status at the time of assessment.
- Table 2 does need some work, in that it needs to be made clearer how many patients developed peroneal nerve palsy following BS within the larger patient cohorts -ie for references 25 and 28. This is over and above what has been stated in point number 4 above.
We have added more details regarding these aspects in Table S2 in the supplementary data section.

Reviewer 2 Report
Comments and Suggestions for Authors
In the present case report and systematic review, Cucu et al. provide a case study of a patient who developed bilateral common peroneal nerve palsy upon bariatric surgery and recovered from this condition through clinical treatment. This case study is complemented with a systematic review of eighteen studies of related cases.
The present work is informative and relevant to clinicians and researchers in its field. It is correctly reported and discussed. Its limitations are indicated. Nevertheless, data presentation can still be improved, as listed below:
1. In Figure 1, please indicate the disc protrusions at L4-L5 and L5-S1 levels with arrows or other signs.
2. In the legend of Figure 3, please indicate the meaning of the arrows, highlighted areas, and dashed line.
3. In Table 1, please move the abbreviations list to a footnote under the table.
4. In the PRISMA 2020 flow diagram, please recheck for consistency the numbers at the identification stage.
5. If correct, please label the PRISMA 2020 checklist as Table S1.
6. In Table S2, please include its title at the top, with the abbreviations list as a footnote.
7. Please mention video S1 in the main text.
Author Response
Dear reviewer, first of all, we would like to thank you for all your comments and suggestions! We believe that due to these suggestions and comments, the quality of our article has increased considerably! The answers and suggested modifications are provided in the main document.
- In Figure 1, please indicate the disc protrusions at L4-L5 and L5-S1 levels with arrows or other signs.
We have fixed this issue. Thank you for the comment!
- In the legend of Figure 3, please indicate the meaning of the arrows, highlighted areas, and dashed line.
We have fixed this issue. Thank you for the comment!
- In Table 1, please move the abbreviations list to a footnote under the table.
We have fixed this issue. Thank you for the comment!
- In the PRISMA 2020 flow diagram, please recheck for consistency the numbers at the identification stage.
Taking into account the reviewers' suggestion, we decided to modify the article into a narrative review, thus, all the specific requirements of a systematic review can no longer be fulfilled. Thank you for your suggestion!
- If correct, please label the PRISMA 2020 checklist as Table S1.
Taking into account the reviewers' suggestion, we decided to modify the article into a narrative review, thus, all the specific requirements of a systematic review can no longer be fulfilled. Thank you for your suggestion!
- In Table S2, please include its title at the top, with the abbreviations list as a footnote.
We have fixed this issue. Thank you for the comment!
- Please mention video S1 in the main text.
We have fixed this issue. Thank you for the comment!

Reviewer 3 Report
Comments and Suggestions for Authors
The Authors decided to describe a case report regarding a nerve paralysis resulting from weight loss. The topic is interesting however a similar ase report has been already described by Lee et al (postgrad med 2024) please underline what is new in the presented case. Moreover as this is a systematic review please provide PROSPERO registration number
There are also cases published in 2025 please revise the search strategy.
Please provide a workflow about how the papers were excluded.
Were there any language limitations?
For the systematic review purpose, please provide the risk of bias analysis.
The introduction is too short, please elaborate more on the topic.
Author Response
Dear reviewer, first of all, we would like to thank you for all your comments and suggestions! We believe that due to these suggestions and comments, the quality of our article has increased considerably! The answers and suggested modifications are provided in the main document.
The Authors decided to describe a case report regarding a nerve paralysis resulting from weight loss. The topic is interesting however a similar ase report has been already described by Lee et al (postgrad med 2024) please underline what is new in the presented case. Moreover as this is a systematic review please provide PROSPERO registration number
- There are also cases published in 2025 please revise the search strategy.
We added the articles published in 2025 as you suggested and made the necessary changes in the text. Thank you very much for your suggestion!
- Please provide a workflow about how the papers were excluded.
A graph describing the steps we followed in the selection of articles included in the study can be seen in Figure 4.
- Were there any language limitations?
Yes, as we mentioned already in the text, we have added in this review only articles written in English.
- For the systematic review purpose, please provide the risk of bias analysis.
Taking into account the reviewers' suggestion, we decided to modify the article into a narrative review, thus, all the specific requirements of a systematic review can no longer be fulfilled. Thank you for your suggestion!
- The introduction is too short, please elaborate more on the topic.
As you suggested, I have added additional information in the introduction. Thank you for your suggestion!

Reviewer 4 Report
Comments and Suggestions for Authors
Authors reviewed the articles to “summarize all reported cases of common peroneal nerve paralysis after weight loss,” (L19). However, this article has not fully answered some of the questions due to insufficient description.
First, authors mixed case report and systematic review, and they did not include method section, which lead to difficulty to read this manuscript. For example, the descriptions “To better integrate the diagnostic and therapeutic approaches…the third reviewer intervened in the decision-making process.” (L179-L196) may be method of the systematic review, but they included these descriptions in discussion section. Authors should rewrite the manuscript, including introduction section, method section, result section, and discussion section.
Second, authors suggest that they conducted systematic review, but they did not show terms they used for search of articles and the number of articles by the search. This manuscript may be a narrative review, but they should describe these results, if they suggest that this manuscript was systematic review. Authors should rewrite the manuscript.
Finally, authors described some of sentences without citation or justification as follows; “According to the WHO definition, obesity is the excessive accumulation of fatty tissue which over time leads to the development of pathologies with metabolic determinism. Obesity is on the rise as urbanization grows, with 18% of adults expected to have this disease by 2030.” (L38), “As the focus on weight management grows, awareness of potential complications from rapid weight loss requires careful consideration.” (L44), “The peroneal nerve, which is one of the terminal branches of the sciatic nerve, is superficially located at the neck of the fibula, covered only by skin and subcutaneous tissue. Because of this anatomical position, it is susceptible to various lesions caused by pressure on the bone. Rapid and significant weight loss can lead to a condition called slimmer's paralysis. With weight loss, the fatty tissue around the fibular head also decreases, therefore the nerve loses its protection against the bone.” (L52), “Since this nerve innervates the lower limb, its compression causes peroneal palsy, resulting in paresthesia, sensory deficits, and weakness in the extensors of the toes, ankle dorsiflexors, and foot evertors. The patient presents an abnormal gait, called a steppage gait, consisting of lifting the foot higher than usual when stepping forward to avoid hitting the toes to the ground.” (L59), “We present the case of a 35-year-old female smoker (8 pack year), with a history of bronchial asthma and elevated blood pressure who presented for cardiovascular evaluation 6 months following bariatric surgery.” (L72), “the Medical Research Council (MRC) scale” (L103), “the Deep Tendon Reflex Scale” (L108), “Mononeuropathies are acute/subacute complications, occurring 1 month to 1-1.5 years after surgery.” (L164), “We have included 18 articles” (L201), “one each from Israel, Greece, Malaysia, South Korea, Brazil, five from the USA, three from Turkey, three from Belgium, and two from Spain.” (L202), “In eight of the 18 articles, the participants lost weight through diet, and in two of them, they also included physical activity. In 8 articles, the subjects experienced complications following bariatric surgery,” (L204), “For the others, the presence or absence of diabetes is not specified. Diabetes and smoking are among the possible risk factors, but in many of the patients in the review, these data are missing.” (L228), “When discussing blood tests, in 12 of the 18 articles included in the review, details related to this aspect are presented.” (L293), and “Nerve conduction studies and electromyography were performed in 371 (98.15%) patients, finding demyelination with partial or complete conduction block of the peroneal nerve at the fibular head.” (L305), but it is difficult for readers to judge them without references as evidence for each description. Authors should add references for these descriptions.
Minor comment
L291: “US” is used without explanation of the abbreviation.
Author Response
Dear reviewer, first of all, we would like to thank you for all your comments and suggestions! We believe that due to these suggestions and comments, the quality of our article has increased considerably! The answers and suggested modifications are provided in the main document.
Authors reviewed the articles to “summarize all reported cases of common peroneal nerve paralysis after weight loss,” (L19). However, this article has not fully answered some of the questions due to insufficient description.
1. First, authors mixed case report and systematic review, and they did not include method section, which lead to difficulty to read this manuscript. For example, the descriptions “To better integrate the diagnostic and therapeutic approaches…the third reviewer intervened in the decision-making process.” (L179-L196) may be method of the systematic review, but they included these descriptions in discussion section. Authors should rewrite the manuscript, including introduction section, method section, result section, and discussion section.
Taking into account the reviewers' suggestion, we decided to modify the article into a narrative review, thus, all the specific requirements of a systematic review can no longer be fulfilled. Moreover, we have modified the narrative review section as you mentioned. Thank you for your comments.
2. Second, authors suggest that they conducted systematic review, but they did not show terms they used for search of articles and the number of articles by the search. This manuscript may be a narrative review, but they should describe these results, if they suggest that this manuscript was systematic review. Authors should rewrite the manuscript.
The description of the selection process of the articles included in the study can be read in the methods section, but also in Figure 4.
Finally, authors described some of sentences without citation or justification as follows; “According to the WHO definition, obesity is the excessive accumulation of fatty tissue which over time leads to the development of pathologies with metabolic determinism. Obesity is on the rise as urbanization grows, with 18% of adults expected to have this disease by 2030.” (L38), “As the focus on weight management grows, awareness of potential complications from rapid weight loss requires careful consideration.” (L44), “The peroneal nerve, which is one of the terminal branches of the sciatic nerve, is superficially located at the neck of the fibula, covered only by skin and subcutaneous tissue. Because of this anatomical position, it is susceptible to various lesions caused by pressure on the bone. Rapid and significant weight loss can lead to a condition called slimmer's paralysis. With weight loss, the fatty tissue around the fibular head also decreases, therefore the nerve loses its protection against the bone.” (L52), “Since this nerve innervates the lower limb, its compression causes peroneal palsy, resulting in paresthesia, sensory deficits, and weakness in the extensors of the toes, ankle dorsiflexors, and foot evertors. The patient presents an abnormal gait, called a steppage gait, consisting of lifting the foot higher than usual when stepping forward to avoid hitting the toes to the ground.” (L59), “We present the case of a 35-year-old female smoker (8 pack year), with a history of bronchial asthma and elevated blood pressure who presented for cardiovascular evaluation 6 months following bariatric surgery.” (L72), “the Medical Research Council (MRC) scale” (L103), “the Deep Tendon Reflex Scale” (L108), “Mononeuropathies are acute/subacute complications, occurring 1 month to 1-1.5 years after surgery.” (L164), “We have included 18 articles” (L201), “one each from Israel, Greece, Malaysia, South Korea, Brazil, five from the USA, three from Turkey, three from Belgium, and two from Spain.” (L202), “In eight of the 18 articles, the participants lost weight through diet, and in two of them, they also included physical activity. In 8 articles, the subjects experienced complications following bariatric surgery,” (L204), “For the others, the presence or absence of diabetes is not specified. Diabetes and smoking are among the possible risk factors, but in many of the patients in the review, these data are missing.” (L228), “When discussing blood tests, in 12 of the 18 articles included in the review, details related to this aspect are presented.” (L293), and “Nerve conduction studies and electromyography were performed in 371 (98.15%) patients, finding demyelination with partial or complete conduction block of the peroneal nerve at the fibular head.” (L305), but it is difficult for readers to judge them without references as evidence for each description. Authors should add references for these descriptions.
We have added more details on the issues mentioned in the introduction and the case report and review section. Furthermore, the references requested in the text have been added. Thanks for your comments. Your suggestions certainly helped us to improve the article!
Minor comment
L291: “US” is used without explanation of the abbreviation.
We have fixed this issue. Thank you for your observation!

Round 2
Reviewer 1 Report
Comments and Suggestions for Authors
Note has been made of the revisions to the manuscript that have been undertaken in response to the previous comments made by the reviewer.
This in itself seems to have introduced some new issues, particularly when it comes to the formatting of the manuscript. There needs to be one Results section along with one Discussion section (currently there are two different Discussion sections -Section 2.1 and Section 3.3). Plus, the Methods section needs to go either before the results or after the Discussion section and Conclusions.
In addition it is essential that the index case (which is the subject of the Case report), is Discussed in such a manner that the case is compared and contrasted with the other previously published 86 cases. This is not always apparent throughout the Discussion section.
One minor issue- the references for the selected publications that form part of the review all appear between lines 224-226 of the manuscript, when in fact usually instead what appears at this stage is a summary of the total number of publications, along with how many of them were either prospective series, case series or case reports. What then is evident is that all of the same publications are then referenced again between lines 227-239 of the manuscript. This section needs further attention.
Author Response
Esteemed reviewer, firstly, we wish to express our gratitude for your comments and suggestions. We are convinced that due to your observations, our article has improved.
Note has been made of the revisions to the manuscript that have been undertaken in response to the previous comments made by the reviewer.
This in itself seems to have introduced some new issues, particularly when it comes to the formatting of the manuscript. There needs to be one Results section along with one Discussion section (currently there are two different Discussion sections -Section 2.1 and Section 3.3). Plus, the Methods section needs to go either before the results or after the Discussion section and Conclusions.
A: We made the requested changes. Thank you for the comment!
In addition it is essential that the index case (which is the subject of the Case report), is Discussed in such a manner that the case is compared and contrasted with the other previously published 86 cases. This is not always apparent throughout the Discussion section.
A: The requested changes can be seen at lines no. 291, 352, 364, 467.
One minor issue- the references for the selected publications that form part of the review all appear between lines 224-226 of the manuscript, when in fact usually instead what appears at this stage is a summary of the total number of publications, along with how many of them were either prospective series, case series or case reports. What then is evident is that all of the same publications are then referenced again between lines 227-239 of the manuscript. This section needs further attention.
A: We made the requested changes, and they can be seen at lines 241-245.

Reviewer 4 Report
Comments and Suggestions for Authors
Authors revised the manuscript, but this manuscript may need some restructuring.
In fact, “3.3. Discussions” may not be the discussion part of “3. Literature review”, but rather that of the entire manuscript. Moreover, “Risk factors”, “Proposed pathophysiological mechanisms”, “Clinical manifestation”, “Investigations”, “Treatment”, and “Study limitations” may be part of the discussion part.
Author Response
Dear reviewer, We have made the structural modifications to the article as you suggested. Thank you for the advice given. Thanks to your suggestions, our article has greatly improved!